# CHATThero: A Language Agent for Recovery Support

## ABSTRACT

Substance use disorders (SUDs) affect millions of people, and relapses are common, requiring multi-session treatments. Access to care is limited, which contributes to the challenge of recovery support. We present **ChatThero**, an innovative low-cost, multi-session, stressor-aware, and memory-persistent autonomous *language agent* designed to facilitate long-term behavior change and therapeutic support in addiction recovery. Unlike existing work that mostly finetuned large language models (LLMs) on patient-therapist conversation data, ChatThero was trained in a multi-agent simulated environment that mirrors real therapy. We created anonymized patient profiles from recovery communities (e.g., Reddit). We classify patients as `easy`, `medium`, and `difficult`, three scales representing their resistance to recovery. We created an external environment by introducing stressors (e.g., social determinants of health) to simulate real-world situations. We dynamically inject clinically-grounded therapeutic strategies (motivational interview and cognitive behavioral therapy). Our evaluation, conducted by both human (blinded clinicians) and LLM-as-Judge, shows that ChatThero is superior in empathy and clinical relevance. We show that stressor simulation improves robustness of ChatThero. Explicit stressors increase relapse-like setbacks, matching real-world patterns. We evaluate ChatThero with behavioral change metrics. On a 1–5 scale, ChatThero raises `motivation` by +1.71 points (from 2.39 to 4.10) and `confidence` by +1.67 points (from 1.52 to 3.19), substantially outperforming GPT-5. On `difficult` patients, ChatThero reaches the success milestone with 26% fewer turns than GPT-5. [1]

## 1 INTRODUCTION

In the United States, about 2.5 million people live with opioid use disorders, and more than 66 million report recent illicit drug use (Recovery Team, 2023). Without sustained support, relapse rates can reach 80–90% in the first year (Wooten & Pieper). At the same time, psychological stress and service demand rise, while access remains scarce (Samji et al., 2022; Grant et al., 2018). Treatments such as medication assisted treatment (MAT), motivational interviewing (MI), and cognitive behavioral therapy (CBT) are effective, yet fewer than one quarter of people with OUD receive care (Cloyes et al., 2010). Barriers include stigma, cost, limited access, and low engagement (Sinha, 2011; Cloyes et al., 2010). Single-session interventions rarely maintain gains; multi-session treatment and continuing care support change and maintenance (Volkow, 2011; Proctor & Herschman, 2014). In counseling, CBT and MI are structured, staged, and span multiple sessions; counselors track state between sessions and reset goals dynamically (Dobson & Dozois, 2021; Hayes & Hofmann, 2018; Fenn & Byrne, 2013).

Recent work advances theory-grounded counseling dialogues, and there is growing interest in multi-turn evaluation (Lee et al., 2024b; Na, 2024; Qiu et al., 2023). Yet most systems were finetuned on single-session or short context, which does not represent real therapeutic process for recovery. Most evaluations use single-turn quality scores, which miss trajectories and relapse-like setbacks (Li et al., 2025; Zhang et al., 2024b; Hu et al., 2024; Qiu et al., 2023; Sun et al., 2021). In this study, we create a multi-session framework to simulate coherent therapeutic strategies under environmental perturbations and reports cross-session outcomes.

---

[1] Code:https://anonymous.4open.science/r/ChatThero-E6E1/README.md

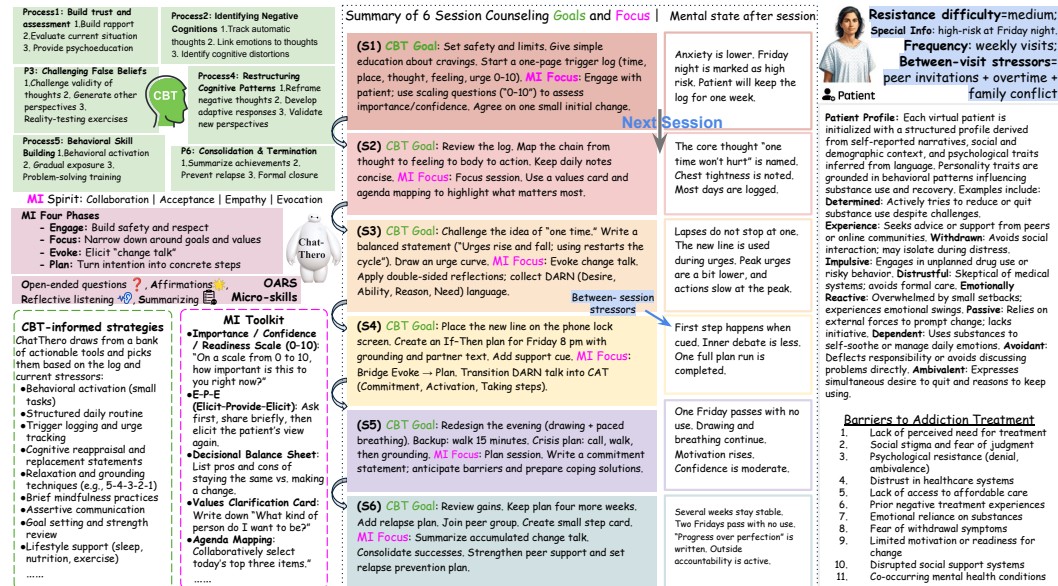

Figure 1: Overview of CHATTHERO. **Left.** Multi-stage CBT with MI. The panel shows CBT stages and a parallel MI layer (Spirit; Four Phases—Engage, Focus, Evoke, Plan; OARS micro-skills; core scales and tools). The Therapy Agent picks from a small bank of CBT strategies and MI tools (0–10 scales, E–P–E, decisional balance, values card, agenda mapping) and then combines and reorders them based on the log and current stressors. Stage tags are for orientation purposes, not in a fixed order. The agent infers the current CBT stage and MI phase and sets the next goal. **Center.** A six-session course shows step-by-step progress. Each session card has a *CBT Goal* and an *MI Focus* that matches the phase (e.g., S1: Engage with scaling; S3: Evoke with double-sided reflections; S5: Plan with commitment and barrier planning). Between sessions, an Environment Agent writes stressors, and the Therapy Agent keeps memory and updates the plan. **Right.** Patient state and profile. Each AI patient starts with a structured Profile (traits, use history, barriers) and a dynamic Memory.

As shown in Figure 1, we frame recovery support as a multi-session, partially observed decision and generation problem with between-session perturbations and craving. We built a simulation based on a realistic environment. (1) A *Patient Agent* holds a persistent state and resistance level (easy/medium/hard); profiles reflect distributions from recovery communities after careful filtering and anonymization (Valdez & Patterson, 2022). (2) An *Environment Agent* injects explicit stressors between sessions to shift state (Sinha et al., 2024). (3) A trainable *Therapy Agent* (ChatThero) chooses and sequences MI/CBT and harm-reduction strategies and keeps memory across sessions (Miller, 1992; McHugh et al., 2010). Training has two stages. First, SFT on clinician-guided synthetic multi-session dialogues learns structure and safety. Second, DPO to decide when to probe, affirm, plan, or switch strategies, with a hard-case curriculum to strengthen carryover under dense stressors.

We evaluate simulated outcomes across sessions. Because MI conceptualizes readiness to change with motivation/confidence rulers and explicitly targets self-efficacy, motivation and confidence are clinically meaningful proximal outcomes for substance-use treatment (of Health et al., 2019). We therefore measure: (1) *motivation* and *confidence* trajectories; (2) *Time-to-Success* (turns to reach a predefined milestone), with emphasis on high-resistance patients; (3) robustness under explicit stressor injections; and (4) agreement between a rubric-anchored LLM-as-Judge and blinded clinicians. In our study, ChatThero improves motivation from 2.39 to 4.10 and confidence from 1.52 to 3.19, and it reduces Time-to-Success on hard cases by 26% turns vs. GPT-5, while scoring higher on empathy and clinical relevance.

Our key **contributions** include: 1) A multi-session, stressor-aware, memory-persistent language-agent formulation with a reproducible simulation. 2) A data–environment construction process: patient profiles from recovery forums and explicit stressor processes that shift state. 3) A two-stage training recipe with a hard-case curriculum for multi-session strategy learning (SFT → DPO). 4) An outcomes-focused multi-session evaluation suite, including motivation/confidence trajectories, Time-to-Success, stressor robustness, and human–LLM agreement, with safety procedures.

## 2 RELATED WORKS

**Substance Use Disorders: Treatment Gaps and Behavioral Challenges.** Evidence-based treatments (e.g., MAT, CBT) reduce relapse and improve functioning (SAM, 2023; Wooten & Pieper; McHugh et al., 2010), yet uptake and retention remain low (e.g., only 22.1% with OUD receive care in a year; <50% remain engaged at 6 months) (Cloyes et al., 2010; SAM, 2023). Barriers include stigma, cost, access, and ambivalence (Sinha, 2011; Cloyes et al., 2010), while episodic therapy often misses high-risk moments (Sinha, 2011). We model evolving resistance and motivation via multi-visit, environment-aware simulations to provide continuity between sessions and test strategy effectiveness under changing conditions.

**LLMs and Automated Counseling for Behavioral Health** LLMs have recently demonstrated strong naturalistic interaction capabilities, making them suitable for health communication (Estrellado & Coy, 2025; Laymouna et al., 2024). AI conversational systems, such as Therabot (Bakoyiannis, 2025), can enhance engagement and adherence through personalized, empathetic dialogue (Cai et al., 2023; Gao et al., 2024), while in clinical practice, LLMs already support triage, diagnostic reasoning, and documentation with efficiency gains (Park et al., 2024; Fan et al., 2024).

**Motivational Interviewing (MI) and Cognitive Behavioral Therapy (CBT).** MI Rollnick (2002); Miller & Rose (2009) and CBT Beck (2019) align with Marlatt's relapse-prevention framework Collier & Marlatt (1995); Frances et al. (2005); Marlatt & Donovan (2005). MI emphasizes empathy, discrepancy, rolling with resistance, and self-efficacy, operationalized via open questions, affirmations, reflections, and eliciting change talk. CBT has strong evidence for substance-use treatment, but in-person delivery and workforce training limit scalability; program requirements can also reduce retention Gertner et al. (2022). Existing chatbots rarely implement MI/CBT explicitly; by contrast, *ChatThero* encodes MI/CBT tactics and learns strategy selection via reinforcement learning within a simulated environment.

**Multi-Agent Simulation, Virtual Patients, and Automated Counseling** Multi-agent simulators model patients, clinicians, and environments; systems like AgentClinic, AMIE, AI Hospital, and MedSimAI use such setups for training and education (Park et al., 2024; Schmidgall et al., 2024; Karthikesalingam & Natarajan, 2024; Fan et al., 2024; Yu et al., 2024). Yet most focus on diagnosis or documentation—not relapse-sensitive behavior change—and seldom include explicit stressors, persistent resistance, or validated MI/CBT tactics (Park et al., 2024; Fan et al., 2024). Virtual patients have long supported medical education and now include LLM-based agents capturing social/psychological dynamics (Gordon et al., 2001; Good, 2003; Garrett et al., 2010; Huang et al., 2007; Campillos-Llanos et al., 2021; Park et al., 2023; Lee et al., 2024a; Wang et al., 2024; Park et al., 2024). With real counseling data scarce, recent work turns to synthetic/reconstructed dialogues (PsyQA, SMILE, CBT-LLM, CPsyCoun, HealMe, CACTUS) (Sun et al., 2021; Qiu et al., 2023; Na, 2024; Zhang et al., 2024b; Xiao et al., 2024; Lee et al., 2024a), but typically in single-session, short-context, static-prompt settings (Park et al., 2024; Fan et al., 2024). In contrast, we embed MI/CBT into *multi-session* simulations, add social triggers and stressors, and build addiction-tailored virtual patients with profiles and evolving memory to test strategy choice, carryover, and robustness.

## 3 METHOD

We model *ChatThero* as a *multi-session, multi-agent* system (Figs. 1, 2) with a Patient Agent (persistent state; easy/medium/hard resistance), an Environment Agent injecting between-session stressors, and a trainable Therapy Agent (ChatThero) that selects/sequences MI/CBT/harm-reduction strategies while maintaining longitudinal memory. This setup carries information across visits, adapts plans to adherence signals and stressors, and advances stage-wise goals. The pipeline includes: (1) Patient Profile Generation, (2) Synthetic Data via Multi-Agent Simulation, and (3) Two-Stage Training (SFT → DPO).

### 3.1 PATIENT PROFILE GENERATION

Recent studies demonstrated the potential of LLMs to simulate realistic patient behaviors and psychological profiles Yao & Yu (2025); Louie et al. (2024); Du et al. (2024); Yu et al. (2024); Lim

et al. (2024). Building upon these advances, our patient simulation consists of a structured **Profile** and dynamically evolving **Memory**.

**Structured Profile.** Patient profiles are generated via an ethically-informed, multi-stage pipeline based on publicly available narratives from Reddit communities related to addiction (e.g., `r/leaves`, `r/addiction`). To ensure privacy, we first prompt a local LLM to identify and remove any potentially privacy-sensitive or personally identifiable information (PII) from the original posts. After this privacy filtering step, the model summarizes key behavioral indicators and psychological patterns (e.g., sleep disturbances, social isolation) using structured extraction prompts (see Table 9). Finally, GPT-4o synthesizes a structured profile consisting of: 1) **Personality Traits** (e.g., impulsivity, emotional reactivity); 2) **Substance Use History** (e.g., duration, frequency, relapse history); 3) **Significant Life Events** (e.g., job loss, social isolation); 4) **Hypothesized Motivations for Substance Use** (e.g., stress relief, coping with insomnia). This multi-step approach allows the LLM to generate plausible yet synthetic patient profiles that capture behavioral realism while ensuring that no actual Reddit user's profile is directly reproduced or reused. As a result, the profiles support downstream experimentation while upholding high ethical and privacy standards.

**Dynamic Memory.** Each patient agent maintains a dynamically evolving memory, recording interactions, emotional states, coping mechanisms, and perceived environmental influences (e.g., relationships, peer pressures, life stressors). An environment agent periodically injects external simulated events (e.g., job loss, relationship breakdown) into the patient's memory stream, realistically altering motivations and behaviors.

### 3.2 SINGLE-SESSION SYNTHETIC DATA (AGENTS & OBJECTIVES)

Because real-world addiction-treatment dialogues are scarce and sensitive, we first construct *single-session* synthetic data to teach core therapeutic skills within one encounter (Figure 2, middle).

**Agents and MI/CBT usage (single-session).** We instantiate two agents: (i) a **Patient Agent (PA)** that presents substance-use–specific concerns, goals, and resistance from a structured profile (motivation, confidence/self-efficacy, current craving level, high-risk triggers/contexts, adherence to plans or MOUD, recent lapses) and an explicit difficulty label {easy, medium, hard}. Here, easy denotes patients who are willing to accept treatment and actively cooperate; medium denotes patients who show ambivalence with noticeable resistance to guidance; and hard denotes long-term users who distrust clinicians and struggle to resist drug-use urges. The PA surfaces these states each turn as a compact summary to guide the therapist policy. (ii) A trainable **Therapy Agent(TA)** that conducts an MI/CBT-consistent session by selecting from predefined

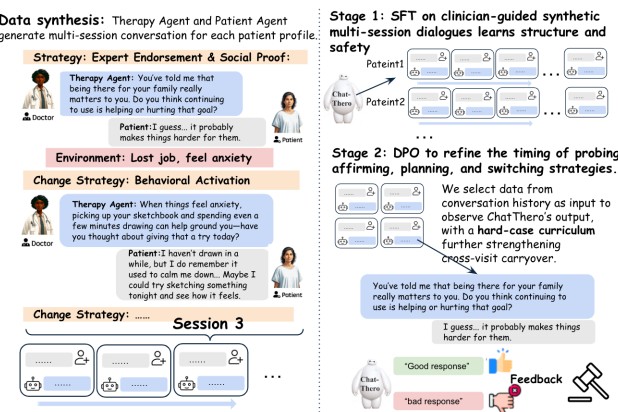

Figure 2: Data synthesis and two-stage training for ChatThero. Left: A Therapy Agent and a Patient Agent generate multi-session dialogues while an Environment Agent injects stressors, prompting strategy switches. Right: Stage-1 SFT teaches safe MI/CBT structure; Stage-2 DPO refines timing and strategy selection. At test time, the agent keeps memory across sessions, adapts after stressors, and targets higher motivation/confidence with lower time-to-success.

strategy primitives (Appendix Table 6) mapped to MI and/or CBT with explicit schemas. Concretely for MI, the policy targets substance-use change by eliciting change talk around personally valued goals (health, legal, family), running decisional balance on use vs. not-use days, using confidence/importance rulers for quit/cut-down plans, rolling with resistance about abstinence vs. harm-reduction, and shaping commitment language for next-step actions. For CBT, the policy operates on drug-use episodes via functional analysis (antecedent–behavior–consequence), trigger mapping (people/places/things), cognitive reappraisal of urges (e.g., "I can't sleep without using"), coping-skill rehearsal for cravings (delay, distract, de-stress, grounding), refusal-skills scripting,

stimulus control (avoiding procurement cues), graded exposure to high-risk cues with urge-surfing, and problem-solving for access/adherence barriers (appointments, transportation).

**Interaction protocol.** Each dialogue follows a substance-use–specific agenda within a fixed-turn budget ($\sim$60 turns): rapport & goal alignment (values, quit/cut-down intent, MOUD preferences) $\rightarrow$ episode clarification (concise functional analysis of a recent use/near-use: antecedents, triggers, craving peak, consequences) $\rightarrow$ plan formulation (translate insights into one or two actionable steps) $\rightarrow$ next-step micro-commitment. In lieu of generic "homework," the session ends with a concrete, time-bounded micro-assignment tied to risk and feasibility—e.g., a trigger/urge log (time, place, people, intensity), a three-line coping card for an anticipated window, a single refusal-line rehearsal with a specific peer, a stimulus-control action (remove a procurement contact/app). The Patient Agent provides a compact state summary each turn (craving, trigger salience, confidence, recent lapse flags), and the Therapy Agent generates the utterance plus a measurable micro-commitment (deadline and success criterion).

**Safety and fidelity controls.** All turns pass rule-based filters (diagnosis prohibition, medication disclaimers, crisis routing) and guardrails to penalize off-policy patterns (e.g., argumentation, prescriptive advice without permission).

### 3.3 Multi-Session Synthesis

To address single-session limitations, we extend synthesis to *multi-session* episodes (3–6 outpatient visits) and explicitly stage therapy across sessions.

**Therapy staging across sessions (strategy evolves).** The TA does not use a static policy; it follows the six CBT stages shown in Fig. 1 (Left) and operationalized in Fig. 1 (Center): **(S1) Build trust & assessment** — set safety/limits and begin a one-page trigger/urge log; MI focus: engage and use 0–10 rulers for importance/confidence. **(S2) Identifying negative cognitions** — review the log and map thought $\rightarrow$ feeling $\rightarrow$ body $\rightarrow$ action; MI focus: values/agenda mapping to sharpen goals. **(S3) Challenging false beliefs** — name the core thought ("one time won't hurt"), draw an urge curve, and reality-test; MI focus: evoke change talk (DARN) with double-sided reflections. **(S4) Restructuring cognitive patterns** — create an If–Then plan with grounding and partner support; MI focus: bridge Evoke $\rightarrow$ Plan (CAT: commitment, activation, taking steps). **(S5) Behavioral skill building** — redesign a high-risk window (e.g., drawing + paced breathing) with a crisis micro-plan and backup steps; MI focus: planning with barrier-anticipation. **(S6) Consolidation & termination** — review gains, extend the plan (four more weeks), add relapse-prevention and peer support; MI focus: summarize accumulated change talk and set maintenance. Across sessions, the PA maintains a persistent memory $m_t$ (craving, trigger salience, motivation, self-efficacy, adherence, lapse flag), summarized to TA at the start of each session so that strategy choice advances the current stage rather than treating sessions as isolated events.

**Environment Agent and rule-based memory updates (Fig. 2).** Between sessions, an **Environment Agent(EA)** injects stressors from a curated catalog (short NL description, severity=*low/medium/high*, duration=days–weeks), organized as: (a) peer/availability (invitations, proximity to using peers, easier access), (b) work/academic (deadlines, shift changes, supervisor pressure), and (c) home/context (family conflict, housing instability, stigma). Each event is recorded in a stressor ledger $\mathcal{L}_t$ and applied to the persistent patient memory $m_t$ via interpretable, rule-based updates—e.g., peer exposure $\uparrow$ trigger salience; sustained workload $\downarrow$ self-efficacy; supportive events or successful refusals $\uparrow$ motivation/confidence; thresholded accumulation toggles lapse_flag. At the next visit, TA receives the $(m_t, \mathcal{L}_t)$ summary, references prior stage artifacts (log, thought record, coping card, plan), and adjusts the staged goal (S1–S6) accordingly—mirroring the "mental state after session" trajectory in Fig. 1 (Right). The whole detailed clinical scenarios settings are shown in Appendix A.2

### 3.4 Training (SFT & DPO): Objectives, Data Construction, and Procedure

**Supervised Fine-Tuning (SFT).** We first supervise ChatThero on single-session dialogues synthesized by GPT-4o from structured patient profiles, using prompts that require MI/CBT/harm-reduction grounding and safety boundaries (Appendix Tables 4, 5); each profile yields multiple scenarios for lexical/situational diversity. The target is the therapist reply $y$ given user turn $x$ and the PA state summary $s$, including an internal strategy tag/rationale (hidden at inference). We then extend SFT to

multi-session episodes (3–6 sessions) that explicitly reference prior plans/outcomes and update state under between-visit stressors; inputs are augmented to $(x, s, m_t, \mathcal{L}_t)$ so the model learns cross-visit carryover (artifact updates, plan continuation). Training uses standard token-level cross-entropy, mixing single-session and multi-session examples in one curriculum.

**Direct Preference Optimization (DPO).** For single-session preference learning, a lightweight simulation pairs a PA (GPT-4o-mini) expressing profile-aligned resistance with candidate TA replies drawn from 18 predefined MI/CBT strategy primitives (Appendix Table 6). Candidates are ranked by GPT-4o using a rubric (empathy, clinical relevance, safety, concreteness), producing preference pairs (Appendix Table 10); we optimize a standard DPO objective with the SFT policy as reference. For multi-session DPO, we form longitudinal preference pairs at cross-visit decision points (relapse framing, strategy switching, plan revision under new stressors, correct use of prior information) and expand the rubric with carryover fidelity, stressor alignment, and stage coherence (Appendix Table 12); these items are up-weighted when forming preferences. This two-phase (single-session $\rightarrow$ multi-session) DPO improves within-visit quality and multi-session persistence without requiring an explicit reward model. [2]

## 4 EXPERIMENTAL SETUP

### 4.1 DATASET

We constructed a large-scale synthetic dataset to simulate realistic therapeutic dialogues related to substance use and recovery. The initial data comprise anonymized and carefully filtered Reddit posts from addiction recovery communities (`r/leaves`, `r/addiction`), involving 57,471 unique authors, with an average of 18.25 posts per author, and 2.13 main posts explicitly addressing substance use challenges. To protect privacy and adhere to ethical standards, all personally identifiable information (PII) was rigorously removed and verified via manual checks.

The resulting synthetic dataset contains 60,471 dialogues, generated through our multi-agent simulation approach. Each synthetic conversation has an average of 45.72 dialogue turns, intentionally matching typical clinical session lengths documented in motivational interviewing (MI) literature Hettema et al. (2005) (see statistics in Appendix Table 2).

Beyond single-session simulations, we construct *multi-visit* trajectories to capture relapse and recovery dynamics over time. Each trajectory comprises 3–6 visits. Between visits, an Environment Agent injects exogenous stressors (e.g., peer use/invitations, access/availability cues, work/academic pressure, family conflict) that update a persistent patient memory (triggers, craving, motivation, self-efficacy, adherence). At each new visit, the Therapy Agent conditions on this carryover state to select strategies (MI/CBT/harm reduction), and the Patient Agent's resistance level (Easy/Medium/Hard) modulates responsiveness. We log visit-level outcomes (abstinent vs. relapse since last visit), auxiliary self-reports (days sober, perceived craving/motivation changes), and a process label indicating whether predefined success criteria were met. The multi-visit subset includes 8,240 dialogues (six-visit arcs), enabling longitudinal evaluation of strategy carryover and environment-induced setbacks.

### 4.2 MODEL BASELINE

We evaluated persuasion capabilities across several state-of-the-art LLMs: GPT-4o, GPT-4o-mini, LLaMA3.1-8B-Instruct, and Qwen2.5 models (7B, 14B, 32B). To ensure a fair and standardized comparison, all models received identical domain-specific prompts that explicitly incorporate therapeutic strategies (MI, CBT, and harm reduction techniques) (Appendix 8). Decoding temperature was set consistently at 0.7 across all evaluations.

### 4.3 AUTOMATIC EVALUATION

**Outcomes.** We score two patient-centered outcomes that are central to behavior change: *Motivation* (readiness/commitment to refrain from use and to engage in coping plans) and *Confidence* (self-

---

[2] We filter generations with safety/ethics rules, retain only privacy-preserving synthesized profiles, and de-duplicate near-duplicates. Hyperparameters and sampling/ranking templates are detailed in Appendix 8–12.

efficacy to resist cues and sustain the plan). Both are rated on a 1–5 Likert scale with anchored descriptors.

**Scoring method and reliability.** A constrained, rubric-anchored LLM scorer (GPT-4o) reads the dialogue state and outputs structured JSON with a numeric score and evidence snippets (rubric/prompts in Appendix A.5). To support reliability, we (i) use anchor descriptors for each point on the scale, (ii) require evidence citations (quoted spans) for each judgment, and (iii) report alignment with human ratings on a stratified sample (rank correlations and calibration discussed in the Appendix). We aggregate across seed reruns via medians to reduce rater variance.

**Single-session reporting.** For each case we compute the *start* score (first turn), *end* score (last turn). We summarize by difficulty (Easy/Medium/Hard) and model.

**Multi-session reporting.** For longitudinal episodes (3–6 sessions), we compute *start*/*end* per session and visualize trajectories across sessions (e.g., grouped, stacked bars). We also summarize per-model, per-difficulty aggregates over episodes.

**Process metric: Time-to-Success (%↓).** We additionally report the percentage of total conversational turns (or sessions) elapsed when the patient first meets a predefined success criterion (e.g., Motivation $\geq 4/5$). This metric captures *how quickly* commitment is achieved (speed matters for adherence and attrition) and is sensitive to environmental setbacks between sessions. We report Time-to-Success in both single-session and multi-session settings alongside outcome scores.

### 4.4 HUMAN EVALUATION

**Rubric and protocol.** Two licensed physicians served as expert annotators.[3] Raters act as clinical conversation assessors and assign 1–5 scores (decimals allowed) on five dimensions:

1) **Responsiveness (R)**: how effectively the clinician addresses the patient's concerns, emotions, and questions; 2) **Empathy (E)**: the degree of emotional sensitivity and support conveyed by the clinician; 3) **Persuasive Strategy Appropriateness (P)**: whether chosen strategies (evidence-based reasoning, analogies, motivational interviewing) match the patient's resistance level; 4) **Clinical Relevance (C)**: accuracy and therapeutic validity within a substance-use context; 5) **Behavioral Realism (B)**: whether interaction style, tone, and pacing resemble real-world clinician behavior.

**Pairwise preference and reporting.** Experts also conduct *comparative* judgments: for matched prompts, they compare our model's response against a baseline and indicate a preference, yielding a win rate (pairwise preference). Raters are blinded to model identity; items are randomized and balanced across conditions. We report per-dimension means and confidence intervals, inter-rater agreement, and alignment with the automatic scorer (details in Appendix A.5). Time-to-Success is reported alongside rubric scores to jointly reflect *quality* (what is said) and *efficiency* (how quickly commitment is achieved).

## 5 RESULTS AND DISCUSSION

We report results around three research questions (RQ1–RQ3) and then provide ablations and supporting analyses. Unless noted, outcome metrics are the rubric-anchored *motivation* and *confidence* scores (1–5; higher is better), and the process metric is *time-to-success* (fraction of turns until first reaching the success threshold; lower is better). Human ratings are by licensed clinicians; automatic ratings use an LLM judge with evidence-cited justifications.

### 5.1 RQ1: HOW DO OUT-OF-THE-BOX GPT/QWEN MODELS BEHAVE ACROSS SETTINGS?

**Single-session (Easy/Medium/Hard).** In *easy* cases, GPT-5, GPT-4o, GPT-4o-mini, Qwen2.5, and LLaMA3.1-8B achieve similar end-of-visit outcomes, indicating a ceiling effect when baseline motivation is high and dependence is mild. In *medium* and *hard* cases, general-purpose models yield modest gains and then plateau; in medium cases they frequently stabilize around end $\approx 2.2$–$2.8$,

---

[3]Written consent was obtained prior to participation; each annotator received a $50 gift card as compensation. More details are provided in Appendix A.5.

suggesting limited ability to move patients past commitment within a single encounter. These observations establish a clear performance gap precisely where persuasive counseling is most needed.

**Multi-session trajectories.** Figure 4 summarizes six-visit trajectories using grouped, stacked bars. For *easy* patients, all models approach the ceiling by visit 1. For *medium* patients, general models improve slowly across sessions and often fail to cross clinically meaningful thresholds, while our domain-aligned agent exhibits larger within-visit lifts *and* higher start scores over time, indicating better cross-visit carryover. For *hard* patients, all systems show in-session gains but weak retention between sessions under realistic triggers; relapse and motivational decay remain common despite transient improvements.

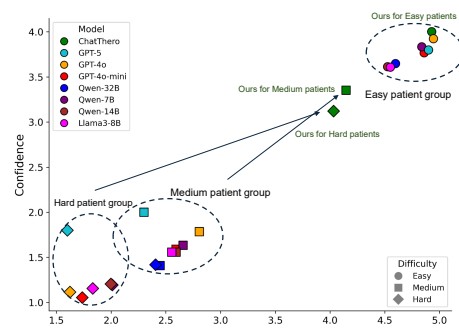

Figure 3: Single-session, no between-session stressors. Each point is the *end-of-visit* Motivation/Confidence mean (1–5). Colors denote models; shapes denote difficulty (Easy/Medium/Hard). ChatThero (based on Qwen-7B) scores highest across all difficulties, with the largest margins on Medium/Hard cases.

**Why fail?** Qualitative inspection of baseline outputs reveals three recurrent issues that align with lower human scores in Table 1: (i) reliance on generic reassurance rather than MI/CBT micro-skills (e.g., reflective listening, decisional balance), (ii) limited handling of resistance and missed opportunities to elicit change talk, and (iii) action plans that fail to incorporate the most recent lapse or newly surfaced stressors. Open-source baselines also require more turns, consistent with higher time-to-success (72–79%).

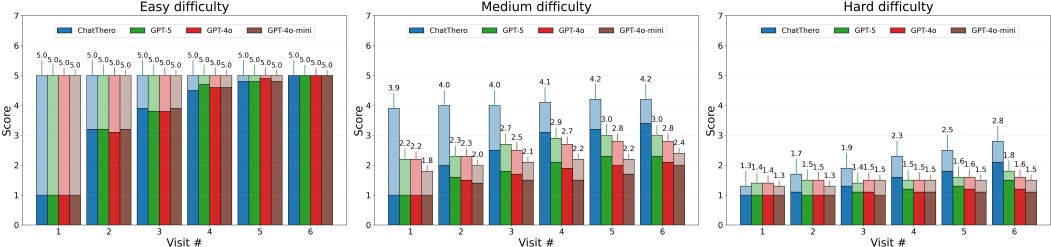

Figure 4: Six-visit episodes with between-session stressors. Grouped, stacked bars by difficulty (Easy/Medium/Hard). Within each visit, bars show *ChatThero*, *GPT-4o*, and *GPT-4o-mini*. Dark segment = start score; light segment = within-visit gain; numbers above bars = end score (1–5). ChatThero shows larger within-visit lifts and higher start scores over time, indicating better carryover under stress, especially on Medium/Hard patients.

## 5.2 RQ2: How does our method improve the agent in single-session settings?

**Outcomes and efficiency.** Starting from Qwen, supervised fine-tuning (FT) improves safety and structure, and Direct Preference Optimization (DPO) yields additional gains. In Table 1, the DPO variant achieves higher empathy (4.93), strategy appropriateness (4.75), and behavioral realism (4.69) than baselines, while reducing time-to-success to 26% of turns (vs. 54% for GPT-4o, 62% for GPT-4o-mini, and 72–79% for open-source baselines). Human ratings align with the LLM judge, supporting reliability.

**Mechanism of improvement.** FT instills a safe conversational scaffold (goal-setting, reflective listening, safety language). DPO further calibrates *when* to deploy which tactic (e.g., timely shift from information-

|  | R | E | P | C | B | W | H | T (↓) |
|---|---|---|---|---|---|---|---|---|
| GPT-4o | 4.68 | 4.87 | 4.39 | 4.47 | 4.50 | 65.5% | 62% | 54% |
| GPT-4o mini | 4.66 | 4.86 | 4.38 | 4.49 | 4.46 | 69.4% | 71% | 62% |
| Qwen2.5-7B | 4.33 | 4.58 | 4.02 | 4.53 | 4.24 | 85.3% | 90% | 79% |
| Qwen2.5-14B | 4.52 | 4.58 | 4.28 | 4.56 | 4.38 | 83.3% | 89% | 74% |
| Qwen2.5-32B | 4.53 | 4.56 | 4.25 | 4.60 | 4.36 | 82.4% | 85% | 72% |
| LLaMA3.1-8B | 4.43 | 4.62 | 4.13 | 4.58 | 4.34 | 82.0% | 91% | 78% |
| ChatThero-FT | **4.81** | **4.90** | 4.66 | 4.58 | **4.65** | - | - | 48% |
| ChatThero-DPO | **4.85** | **4.93** | **4.75** | **4.61** | **4.69** | **42.3%** | 41% | 26% |

Table 1: single-session, standardized 60-turn cap. R/E/P/C/B are clinician-style ratings (1–5; higher is better). **W** = pairwise win rate vs. ChatThero-FT judged by a rubric-anchored GPT-4o. **H** = share of dialogues scored by blinded clinicians. **Time-to-Success** = fraction of turns when the success threshold (Motivation ≥ 4/5) is first reached; lower is better.

giving to decisional balance; proactive refusal-skills rehearsal when cue salience is high). As a result, the policy resolves more medium and hard cases earlier, consistent with shorter trajectories and higher end-of-visit scores (cf. Fig. 4). Ablations in Table 1 confirm a two-stage effect: FT raises safety and structural coherence; DPO adds adaptive control of tactic selection. Relative to base Qwen, both outcomes and efficiency improve; relative to FT, DPO yields further gains, especially in medium/hard cases where timing and tactic choice are critical.

**Strategy diversity vs. outcomes.** Simply using more strategies is not sufficient. The right panel of Fig. 5 shows a stronger positive association between unique strategies and outcomes for our model, particularly in hard settings, whereas baselines exhibit weak or negative associations. The benefit arises from *context-sensitive deployment*, not variety per se.

### 5.3 RQ3: How does the chatbot perform in multi-session settings with social stressors, and what remains difficult?

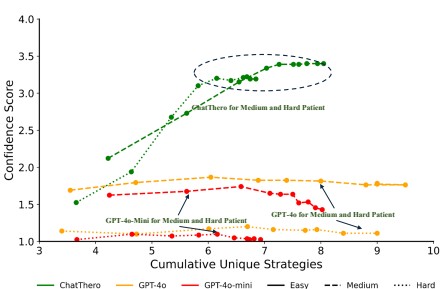

Figure 5: single-session, per-difficulty analysis. X-axis = number of unique persuasive strategies used in a dialogue (from the predefined pool in Table 6); Y-axis = resulting Motivation (1–5). Lines denote difficulty (solid/dashed/dotted). ChatThero shows a stronger positive relation in Medium/Hard cases, suggesting gains come from context-sensitive deployment rather than variety alone. Full information for both motivation and confidence can be found in Figure 7.

**Overall behavior under stress.** With stressors injected between sessions (peer exposure, workload, family/environment), our agent exhibits the strongest aggregate trend: larger within-visit improvements and upward drift in start scores for *medium* cases, and the highest end-of-visit scores by visit 6 for *hard* cases (though margins remain modest). This suggests partial robustness to exogenous pressures.

**Unresolved failure modes.** Even with memory carryover and a stressor log, three problems persist: (1) refusal-skill practice does not start early enough before known high-risk periods; (2) strategy switching is slow after new setbacks; (3) safety planning often comes late within a visit. These issues align with loss of gains between sessions and renewed ambivalence in hard cases. This points to a need for learning signals that reflect long-term goals, not only within-visit progress.

A responsible next step is to frame this as an open challenge for the community and to study methods that can hold gains across sessions under realistic stress. In particular, work that (i) **designs environments** that are partially observable and ledger-driven, where peer exposure, workload, and home stress enter as outside events, and where adherence, relapse, and skill practice are state variables; (ii) **uses outcome-anchored rewards** that go beyond turn-level proxies and instead track longer signals tied to abstinence, care-plan adherence, and timely use of coping skills, while mixing sparse success signals with shaped intermediate steps (e.g., plan execution, craving control) for credit assignment; (iii) **learns reward models from preferences and feedback** collected as structured clinician ratings and evidence-cited LLM critiques, then aggregates short-horizon conversation quality with longer clinical trajectories using simple time aggregation and objectives that account for risk; and (iv) **tests planning and safety** with model-based or hierarchical RL that schedules proactive actions (for example, refusal-skill rehearsal before predicted high-risk intervals) while enforcing safety rules and de-escalation, may provide clearer evidence on what does and does not transfer to multi-session care.

## 6 Conclusion

This work presents CHATTHERO, a domain-aligned conversational agent for addiction recovery. Across single- and multi-session settings, CHATTHERO outperforms strong general models on motivation/confidence outcomes and time-to-success, and receives higher human ratings for empathy, strategy appropriateness, and behavioral realism. The gains are pronounced in medium and hard cases, where generic systems plateau.

LIMITATIONS AND ETHICAL STATEMENT

Despite promising gains in AI-assisted interventions in addiction recovery, several limitations remain. First, our patient simulations—based on structured profiles synthesized from anonymized Reddit narratives and LLM generation—may not fully capture the emotional complexity and interpersonal unpredictability of real-world clinical settings, despite rigorous privacy controls and human validation. Future work should integrate standardized patient actors and real clinical supervision to further enhance realism and robustness. Second, our study is limited to English-language, Western-context scenarios. The persuasive strategies validated here may not generalize across diverse cultural, linguistic, or treatment contexts. Extending to broader populations and non-English environments is essential for ensuring fairness and global relevance. Third, our evaluation focuses on short-term conversational outcomes across key clinical dimensions but does not yet assess longitudinal effects such as patient trust, therapeutic alliance, or sustained behavioral change. Addressing these gaps will require real-world trials and long-term patient-reported outcomes. Fourth, even in multi-session settings that model environmental pressures (e.g., family dynamics, everyday stressors, peers who continue to use), hard patients—often with long use histories—remain difficult to persuade: in-session gains tend to erode between sessions, and progress is fragile, indicating a substantial gap from solving real-world recovery at scale. Bridging this gap likely requires coupling dialog systems with environment-level supports (trigger management, contingency management, family/peer engagement) and integrated clinical care. Finally, our framework focuses on structured, simulated therapeutic dialogue and may not adequately address highly sensitive or ethically complex cases (e.g., trauma, suicidality, acute crises). These require additional safeguards, risk protocols, and close involvement from licensed clinicians before any real-world deployment.

This research advances conversational AI capabilities in the sensitive domain of addiction recovery, necessitating strict adherence to ethical standards and careful risk assessment: Firstly, the dataset underlying our simulation framework was generated exclusively from publicly accessible and anonymized Reddit data. Rigorous steps were taken to remove personally identifiable information (PII), and independent manual audits were conducted to verify anonymity and privacy compliance. We also have IRB to collect these datasets. However, due to the sensitive and potentially identifiable nature of addiction narratives, the dataset and synthetic profiles are not released publicly. Instead, we provide only high-level data statistics, synthetic examples, and detailed methodology for reproducibility. Secondly, the therapeutic strategies incorporated in our model were derived from established psychological frameworks (CBT, MI, harm reduction), with consultation from licensed clinicians ensuring clinical validity. However, the conversations synthesized by the LLMs were not vetted for real-world clinical deployment. Significant risks remain regarding potential misinformation, inappropriate therapeutic recommendations, or inadequate emotional responses if such a system were used without rigorous oversight. Thirdly, all human evaluation procedures were conducted by licensed healthcare professionals who provided explicit informed consent and received fair compensation for their participation. The evaluation was conducted under an academic IRB-exempt protocol strictly following ethical guidelines for responsible human-subjects research. We explicitly caution against deploying ChatThero or similar conversational agents in clinical or therapeutic settings without extensive clinical validation, robust safety frameworks, ongoing expert supervision, and a thorough assessment of potential harms. Interdisciplinary collaboration between AI researchers, clinical experts, ethicists, and patient advocacy groups remains imperative to ensure that persuasive conversational AI tools contribute safely and equitably to the recovery process from addiction.

REPRODUCIBILITY STATEMENT

We provide full details to ensure reproducibility. Dataset sources and splits are in Experiment section; implementation details and training practices are in Experiment section; hyperparameters are listed in Experiment section; compute setup is described in Experiment section. We also include an anonymized code repository link [4].

---

[4]https://anonymous.4open.science/r/ChatThero-E6E1/README.md

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

## A APPENDIX

### A.1 LLM USAGE

In accordance with the ICLR 2026 policies on LLM usage, we disclose how LLMs were used in this work. LLMs were employed to assist with grammar polishing, wording improvements, and drafting text during paper preparation. All technical content, proofs, experiments, and analyses were conceived, implemented, and validated by the authors. Authors remain fully responsible for the correctness of the claims and results.

No LLMs were used to generate research ideas, write code for experiments, or produce results. No confidential information was shared with LLMs, and no prompt injections or other inappropriate uses were involved.

This disclosure aligns with the ICLR Code of Ethics: contributions of tools are acknowledged, while accountability and verification rest entirely with the human authors.

### A.2 CLINICAL SCENARIOS DESIGN

**Difficulty levels and patient prompts**    We partition virtual patients into three levels: Easy, Medium, and Hard, reflecting anticipated persuasion difficulty and severity of substance-use dependence. Difficulty is instantiated through profile-conditioned prompts that specify motivational barriers, prior treatment history, and stance toward change. **Easy**: generally receptive to intervention; expresses willingness to follow coping plans and try alternative behaviors. **Medium**: shows partial resistance, may reject some strategies; requires the therapist agent to adapt and switch tactics dynamically. **Hard**: exhibits entrenched pessimism and low self-efficacy; doubts both personal ability to recover and the effectiveness of therapy. Each of the 300 patients (100 per difficulty level) is initialized with a distinct clinical profile that captures psychological traits, substance-use history, and salient barriers.

**single-session protocol**    We model a standard outpatient session with an upper bound of 60 utterances (approximately 45 minutes, based on clinician role-plays used to calibrate length), providing sufficient headroom for assessment, strategy delivery, planning, and safety language. Dialogues terminate early if the patient indicates resolution (e.g., expresses sufficient motivation and confidence to follow a coping plan). Hitting the 60-utterance cap, therefore, signals unresolved concerns by the end of the session. Standardizing the cap ensures comparability across patients and models. During analysis, we track the start (first turn) and end (last turn) scores for Motivation and Confidence, the within-visit gain $\Delta$, and the distribution/variety of strategies used from the predefined pool (Table 6).

**Multi-session protocol and state carryover**    To capture relapse risk and between-session variability, we extend scenarios to longitudinal episodes with 3–6 sessions. We report six-visit trajectories as the default for clarity and comparability. Six contacts (3–6 months) strike a balance between realism and tractability, and our pilots showed diminishing returns beyond this horizon for most models.

Between sessions, an Environment module samples stressors from a curated catalog (peer/availability, work/academic, home/context; Appendix with short natural-language descriptions, severity (low/medium/high), and duration (days–weeks). These events are written to a stressor ledger. At the next visit, both the patient and therapist agents receive (i) the prior plan/outcomes and (ii) a summarized ledger of intervening stressors.

Patient state is maintained through a lightweight memory that contains motivation, confidence, cravings/trigger salience, self-efficacy, plan adherence, and recent lapses. Memory updates are rule-based: peer exposure increases trigger salience; sustained workload stress reduces self-efficacy and adherence; supportive events or successful refusal attempts raise motivation and confidence. The therapist agent adapts MI/CBT/harm-reduction strategies accordingly (e.g., lapse management, refusal skills, trigger planning), explicitly referencing prior sessions to promote carryover. We evaluate per-visit start/end scores across the six sessions and summarize a process metric, Time-to-Success (the fraction of total turns or sessions elapsed when a success threshold, e.g., Motivation $\geq 4/5$, is first reached; lower is better).

### A.3 PROMPT SETTING

We employ a suite of specialized prompts to facilitate therapeutic conversation generation, simulate resistant patient behavior, conduct multi-dimensional evaluations, and support model comparison. Each prompt is tailored to a distinct task in our framework, and the corresponding templates are provided in Tables 4–11.

**Therapeutic Dialogue Generation (Tables 4, 5, 6)**   This prompt guides the assistant to generate multi-turn, empathetic conversations grounded in a patient's profile analysis. It integrates a diverse range of therapeutic strategies (e.g., MI, CBT, harm reduction), enforces coverage and balance across interventions, and prompts iterative adjustments based on patient reactions. To ensure consistency and clinical utility, the assistant selects from a list of 18 predefined strategies and adheres to behavioral constraints such as natural transitions, reflective validation, and session length (more than 50 turns).

**Conversation Scoring Prompt (Table 7)**   This prompt is used to evaluate generated doctor-patient conversations across five clinically relevant dimensions: Responsiveness, Empathy, Persuasive Strategy Appropriateness, Clinical Relevance, and Behavioral Realism. GPT-4o produces scores in a strict JSON format, enabling structured, scalable, and reproducible comparison of dialogue quality across models.

**Role-Play Simulation Prompts (Table 8)**   We simulate realistic interactions between an assistant and a patient persona (played by GPT-4o-mini) using dual-role prompts. The patient prompt conditions the agent on a full personality profile and a resistance level (Easy, Medium, Hard), while the doctor prompt guides the assistant to respond with strategy, empathy, and adaptiveness. This process produces controlled yet diverse interaction trajectories that can be used for DPO training and case analysis.

**Pairwise Full-Dialogue Comparison Prompt (Table 11)**   To compare the persuasive efficacy and realism of different models, we use a prompt that presents two full conversations and asks GPT-4o to choose the better therapist based solely on their responses. This eliminates confounding influence from patient utterances and focuses evaluation strictly on assistant behavior.

**Human Evaluation and LLM Evaluation**   All participants in this human evaluation were volunteers with at least five years of professional experience in neurology, psychology, or related specialties at hospitals.

**LLM-as-a-Juage**   LLMs are increasingly used as automated judges Li et al. (2024); Gu et al. (2024). Studies show that models like GPT-4 Achiam et al. (2023); Liu et al. (2023); Fu et al. (2023) and critique-tuned variants Ke et al. (2023) can approximate human judgment in summarization Chen et al. (2023), dialogue Zheng et al. (2024); Zhang et al. (2024a), and translation Kocmi & Federmann (2023). In the medical domain, LLM-as-judge has been applied to clinical conversations Tu et al. (2025); Arora et al. (2025); Wang et al. (2023), medical documentation Croxford et al. (2025); Chung et al. (2025); Brake & Schaaf (2024), exam question answering & generation Yao et al. (2024a;b), and medical reasoning Jeong et al. (2024); Tran et al. (2024). Inspired by these works, we introduce a GPT-4o-based LLM-as-a-Judge following the same requirements and settings as the above human evaluation. The details and prompts can be found in Table 6 and 11.

### A.4 TRAJECTORY OF PERSUASION AND EARLY TERMINATION EFFICIENCY

Figure 6 (left panel) further delineates how motivation and confidence evolve throughout dialogues. ChatThero showed rapid improvement and early stabilization, with over 80% of medium and hard scenarios reaching satisfactory resolution within 36 turns, compared to baseline models often unable to resolve these cases effectively. This early resolution efficiency underscores the real-world applicability of our approach, particularly in resource-limited healthcare environments where timely interventions are critical.

### A.5 CONVERSATION QUALITY AND CLINICAL EVALUATION FRAMEWORK

Dialogues were evaluated across five clinically-relevant dimensions—Responsiveness, Empathy, Persuasive Strategy Appropriateness, Clinical Relevance, Behavioral Realism—rated on a 1–5 scale using structured prompts (evaluation prompt detailed in Appendix Table 7). A small-scale human evaluation subset (N=100) further validated synthetic data quality and clinical realism (evaluation results in Appendix Table 3).

Finally, our design addresses ethical concerns explicitly through strict data anonymization, synthesized profiles, and controlled simulations, clearly positioning this study as exploratory. Raw narratives are not released; however, code, data construction scripts, synthetic data examples, and reproducible scenario templates are provided to facilitate ethically responsible follow-up research.

| Statistic | Value |
|---|---|
| #Authors | 57,471 |
| AVG. #Posts Per Author | 18.25 |
| AVG. #Main Posts Per Author | 2.13 |
| #Conversations | 60,471 |
| AVG.#Turns Per Conversation | 45.72 |

Table 2: Descriptive statistics of the collected Reddit-based substance use dataset, including the number of unique authors, average post counts, and dialogue characteristics.

| Model | Empathy | Strategy Use | Clarity |
|---|---|---|---|
| Real-World | 95.0% | 97.0% | 100.0% |
| Synthesis Conversation | 89.0% | 85.0% | 99% |

Table 3: Human Evaluation Results (N=100 dialogues). Each cell reports the proportion of dialogues that **did not exhibit a specific deficiency**, such as lack of empathy, inappropriate strategy use, or unclear expression. Higher values indicate better performance.

Table 4: Prompt Template for Generating Therapeutic Dialogues (Part 1).

**Prompt (Part 1)**

The following is the analysis of a patient:

{user_analysis}

As a therapist meeting this patient for the first time (the doctor didn't have any information of patient to begin with), create a detailed, step-by-step conversation that incorporates the following strategies:

Motivational Interviewing (MI): Explore the individual's values and goals to ignite their motivation for change.

Cognitive Behavioral Therapy (CBT): Identify and modify negative thought patterns and behaviors linked to substance use.

Solution-Focused Brief Therapy (SFBT): Focus on the individual's strengths and past successes to achieve their recovery goals.

Peer Support Programs: Leverage group support or mutual-help networks to foster accountability and a sense of belonging.

Mindfulness-Based Interventions (MBIs): Incorporate mindfulness practices to improve emotional regulation and reduce cravings.

Behavioral Activation (BA): Promote engaging in meaningful activities to replace substance-related behaviors.

Relapse Prevention Strategies: Develop skills to recognize triggers and implement coping mechanisms to avoid relapse.

Strength-Based Approach: Highlight the individual's resilience and personal resources to empower recovery efforts.

Psychoeducation on Addiction and Recovery: Educate the individual about the effects of substances and the benefits of recovery.

Harm Reduction Framework: Provide strategies to minimize immediate harm while working towards cessation.

Family and Social Support Involvement: Engage family or trusted individuals in the process to strengthen the support network.

Self-Compassion Practices: Encourage self-kindness to build confidence and reduce guilt associated with substance use.

Coping Skill Development: Equip the individual with practical skills to manage stress, anxiety, and other challenges without substances.

To ensure balanced use of strategies, here is the current usage count of each strategy: - {strategy_name}: {count} times used. ...

When introducing coping mechanisms or steps for the patient, select from the predefined actionable strategies below:

1. Explore specific hobbies or interests the patient can engage in to replace addictive behaviors (e.g., art, sports, volunteering).

2. Develop a structured daily routine to bring stability and reduce idle time that might trigger relapse.

3. Introduce grounding techniques such as sensory exercises or physical activities to manage anxiety or cravings.

4. Suggest joining a support group or community to build social connections with individuals on similar journeys.

5. Provide psychoeducation on how addiction affects the brain and emotional regulation.

...

18. Support the patient in finding meaningful ways to contribute to their community, such as mentoring, advocacy, or local initiatives, to foster a sense of purpose.

Table 5: Prompt Template for Generating Therapeutic Dialogues (Part 2).

**Prompt (Part 2)**

Ensure the dialogue meets the following requirements: 1. Gradually explore the patient's personality, addiction history, challenges, and triggers through multiple open-ended questions.

2. Use multiple strategies from the above lists throughout the conversation. Avoid defaulting to the same few strategies and instead adapt them to the patient's needs.

...

5. Engage in iterative dialogue for each solution, where the therapist introduces a strategy, seeks the patient's feedback, adjusts based on their response, and explores challenges or barriers before finalizing the approach.

6. Maintain a collaborative and patient-centered approach, where solutions emerge naturally through dialogue rather than being imposed by the therapist.

7. Ensure the conversation spans at least 50 dialogue turns (25 from the therapist and 25 from the patient), reflecting the depth and duration of a real therapeutic session.

8. Use natural transitions to progress from one topic to another, ensuring the conversation feels organic and unhurried.

9. The conversation should begin with the patient's first utterance.

Here is an example of a layered, empathetic dialogue:

**Patient:** Hi... um, thanks for seeing me today. I wasn't sure what to expect.

**Therapist:** Hi Mark, I really appreciate you coming in. Starting this process can feel overwhelming, but I'm here to support you. What's been on your mind lately?

**Patient:** I've been feeling really stuck. I know I want to quit smoking, but every time I try, I just feel like I'm failing all over again.

**Therapist:** I hear you, Mark. Quitting smoking is one of the hardest challenges anyone can take on, and it's completely natural to feel this way. I've worked with others who've felt the same—they described it as climbing a mountain that feels too steep. But I've also seen them reach the top, step by step. Can we talk about what makes the climb feel steep for you right now?

**Patient:** It's the cravings. They just hit me out of nowhere, and I don't know how to handle them.

**Therapist:** Cravings can feel like a storm, can't they? I worked with someone once who described their cravings as waves that kept crashing over them. Together, we found ways for them to ride out those waves, like focusing on a small activity or changing their environment. Could we explore some strategies that might help you ride out your cravings too?

**Patient:** Sure, I guess.

**Therapist:** Great. Let's start with understanding when these cravings hit hardest. For example, is it during specific times of day or situations?

The conversation should continue to explore: - The patient's motivations, barriers, and triggers in detail. - Strategies and coping mechanisms tailored to their unique experiences, ensuring diversity in approaches. - Empathetic reflections from the therapist that validate the patient's feelings and provide relatable examples to instill hope. - Iterative problem-solving where the therapist introduces, discusses, and adjusts strategies collaboratively. - A gradual, layered exploration of the patient's challenges, ensuring at least 50 dialogue turns to reflect the depth of a real therapeutic session.

The goal is to create a natural, empathetic, and multi-layered dialogue that feels authentic and provides actionable, diverse therapeutic strategies. Ensure the length and depth align with the standards of a comprehensive therapy session.

At the end of the conversation, return the strategies used in the following format (must follow the following format like `**Strategies:**`):

`**Strategies:**` Motivational Interviewing (MI), Cognitive Behavioral Therapy (CBT), Peer Support Programs, etc.

Table 6: Predefined Actionable Strategies for Therapeutic Dialogue Generation.

| ID | Strategy Description |
| --- | --- |
| 1 | Explore specific hobbies or interests the patient can engage in to replace addictive behaviors (e.g., art, sports, volunteering). |
| 2 | Develop a structured daily routine to bring stability and reduce idle time that might trigger relapse. |
| 3 | Introduce grounding techniques such as sensory exercises or physical activities to manage anxiety or cravings. |
| 4 | Suggest joining a support group or community to build social connections with individuals on similar journeys. |
| 5 | Provide psychoeducation on how addiction affects the brain and emotional regulation. |
| 6 | Work on identifying and addressing specific emotional triggers through reflective exercises. |
| 7 | Practice assertive communication techniques for setting boundaries with peers or environments that encourage substance use. |
| 8 | Encourage the patient to journal their thoughts and emotions as a way to process experiences and identify patterns related to cravings or triggers. |
| 9 | Introduce relaxation techniques such as progressive muscle relaxation or guided imagery to alleviate stress and improve emotional well-being. |
| 10 | Help the patient set short-term and long-term goals to maintain focus and motivation during their recovery journey. |
| 11 | Explore mindfulness-based activities like meditation, yoga, or tai chi to promote self-awareness and emotional regulation. |
| 12 | Identify and reinforce the patient's personal strengths and past successes to build confidence in their ability to overcome challenges. |
| 13 | Provide education on the importance of nutrition, sleep, and exercise in supporting recovery and overall health. |
| 14 | Develop a crisis plan for managing high-risk situations or moments of intense cravings, including a list of emergency contacts and actions. |
| 15 | Encourage the patient to create a vision board or list of positive outcomes they hope to achieve through recovery as a source of inspiration. |
| 16 | Discuss the concept of gratitude and suggest keeping a gratitude journal to focus on positive aspects of life and maintain perspective. |
| 17 | Offer resources or referrals for complementary therapies, such as art therapy, music therapy, or animal-assisted therapy, to enhance emotional support. |
| 18 | Support the patient in finding meaningful ways to contribute to their community, such as mentoring, advocacy, or local initiatives, to foster a sense of purpose. |

Table 7: Prompt Template for Scoring Doctor-Patient Conversations Across Five Clinical Dimensions.

**Prompt Template:**

You are a professional clinical conversation evaluator. Please assess the quality of the following doctor-patient dialogue across five key dimensions. Assign a rating from 1 to 5 for each criterion (allowing decimals such as 3.7 if appropriate).
Conversation:
{conversation}

Scoring Criteria:
1. Responsiveness (1-5): How well the doctor agent addresses the patient's concerns, emotions, and questions at each turn.
    - 1: Largely ignores or poorly addresses the patient's input
    - 5: Fully acknowledges and appropriately responds to the patient's needs
2. Empathy (1-5): How well the doctor agent shows understanding, compassion, and emotional sensitivity.
    - 1: Shows minimal or no empathy
    - 5: Demonstrates strong emotional support and understanding
3. Persuasive Strategy Appropriateness (1-5): How appropriately the doctor uses persuasive strategies (e.g., evidence-based reasoning, analogies, addressing fears) based on the patient's resistance or concerns.
    - 1: Strategies are inappropriate, irrelevant, or missing
    - 5: Strategies are highly appropriate and tailored to patient needs
4. Clinical Relevance (1-5): How medically accurate and relevant the doctor's responses are regarding the context of drug addiction treatment.
    - 1: Responses are clinically inaccurate or irrelevant
    - 5: Responses are clinically accurate, appropriate, and helpful
5. Behavioral Realism (1-5): How realistically the doctor agent's conversational behavior reflects that of a real-world healthcare professional.
    - 1: Doctor behavior is unnatural or unrealistic
    - 5: Doctor behavior closely mirrors real-world clinical interactions
Please output ONLY the scores in the following JSON format:
{
    "Responsiveness": X,
    "Empathy": X,
    "Persuasive Strategy Appropriateness": X,
    "Clinical Relevance": X,
    "Behavioral Realism": X
}

Table 8: Prompt Templates Used in Role-Play Simulation Between Doctor and Patient.

| Patient Prompt (Role-Played by GPT-4o-mini) | Doctor Prompt (Role-Played by GPT-4o) |
|---|---|
| You need to act as a patient.

Here is your profile:
{analysis}

Conversation history:
{history}

{difficulty description} Now you should continue the conversation with the doctor to explore ways to stop using drugs. If your goal has been achieved, you can say goodbye to the doctor. Otherwise continue to talk with doctor(based on rules).

Just generate the content of the patient's reply only. Don't generate any narration or system messages. Keep the response natural and not too long. | You are a supportive and thoughtful doctor who is helping the patient overcome drug use. Continue the following conversation and provide a helpful, encouraging response that aims to support the patient's recovery.

The strategy you could use:
{strategy}

Example:
Patient: Hi... um, thanks for seeing me today. I wasn't sure what to expect.
Therapist: Hi Mark, I really appreciate you coming in. [...] [...continues with natural empathetic dialogue...]

Conversation history:
{history}

Just generate the doctor's reply only. No narration or tags. |

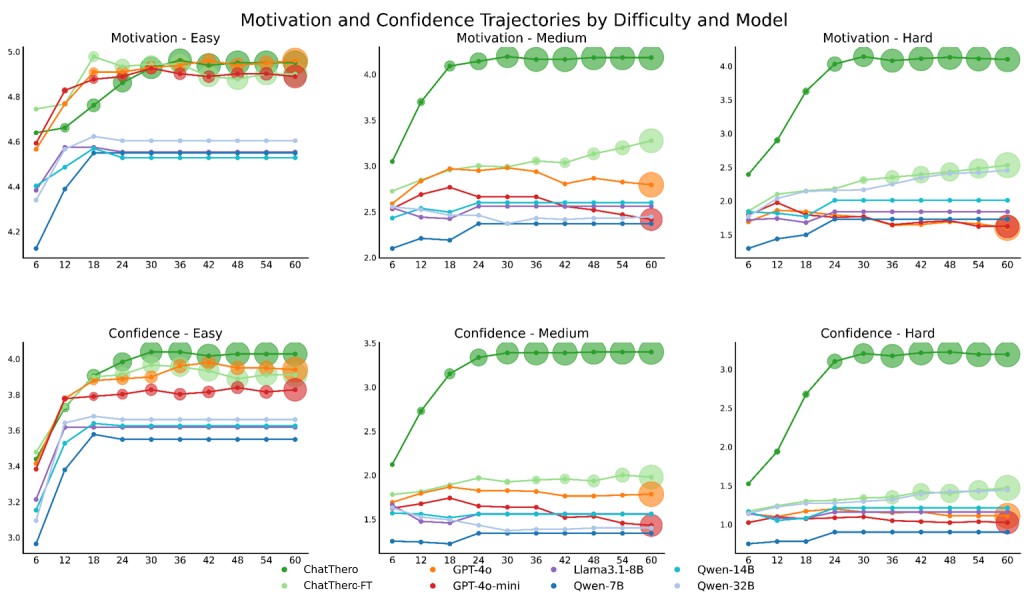

Figure 6: Motivation, Confidence, and the Impact of Strategic Diversity Across Patient Difficulty Levels and Models. Left panel: Motivation and confidence score trajectories over conversation turns for Easy, Medium, and Hard virtual patients, evaluated across all models. The x-axis denotes the number of dialogue turns; the y-axis shows the average motivation (top) or confidence (bottom) score at the final turn of dialogues ending at each point. Each curve corresponds to a different model, with bubble sizes indicating the frequency of dialogue termination at that turn count. Effective models (notably ChatThero) enable more patients to achieve higher motivation and confidence earlier, leading to earlier and more frequent session resolution—especially for harder patients.

Table 9: Prompt Template for Extracting Psychological Traits and Behavioral Themes from Reddit Posts.

---

**Prompt Template:**

---

```
You are a clinical psychology research assistant. Your task is
to read Reddit posts related to addiction and extract structured
information to support the generation of simulated patient
profiles.
Reddit Post:
{reddit_post}

Please extract the following fields from the post content, using
only the information explicitly or implicitly present in the text.
If a field is not mentioned, return \texttt{null}.
1. Personality Traits: Describe any personality characteristics
shown by the poster (e.g., impulsivity, low self-esteem, social
withdrawal).
2. Substance Use History: Summarize type of substance, usage
duration, frequency, withdrawal attempts, or relapse patterns.
3. Significant Life Events: Extract major life events related to
the addiction (e.g., job loss, divorce, trauma, moving cities).
4. Behavioral Themes: Identify relevant behavioral patterns (e.g.,
sleep issues, isolation, risk-taking behavior, dependency).
5. Motivations for Substance Use: Hypothesize possible reasons
for drug use (e.g., coping with anxiety, escaping boredom, peer
pressure).
Please return the extracted fields in the following JSON format:
{
    "Personality Traits": "...",
    "Substance Use History": "...",
    "Significant Life Events": "...",
    "Behavioral Themes": "...",
    "Motivations for Substance Use": "..."
}
```

---

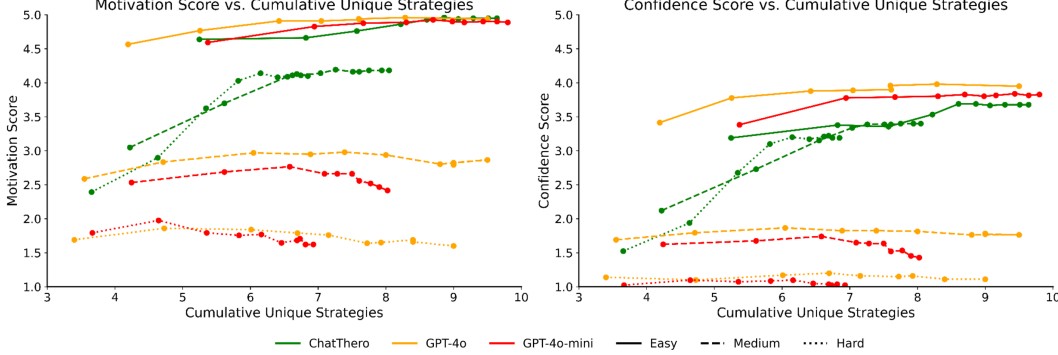

Figure 7: single-session, per-difficulty analysis. X-axis = number of unique persuasive strategies used in a dialogue (from the predefined pool in Table 6); Y-axis = resulting Motivation/Confidence (1–5). Lines denote difficulty (solid/dashed/dotted). ChatThero shows a stronger positive relation in Medium/Hard cases, suggesting gains come from context-sensitive deployment rather than variety alone.

Table 10: Ranking and Annotation Protocol for Therapy Response Selection.

**Protocol:**

You are an expert evaluator (either GPT-4o or a licensed clinical
professional) tasked with ranking multiple candidate therapy responses
generated for a given patient-therapist dialogue state. Please follow
the steps below:
Dialogue State:
{current_dialogue_context}

Candidate Responses:
1. {response_1}
2. {response_2}
3. {response_3}
...

**Ranking Criteria:**
1. Clinical Appropriateness: Assess whether the response is medically
and therapeutically sound, and aligns with evidence-based practices.
2. Empathy and Emotional Support: Evaluate how well the response shows
understanding, compassion, and validation of the patient's feelings.
3. Relevance to Patient Context: Check if the response directly
addresses the patient's current needs, concerns, and emotional state.
4. Clarity and Communication Style: Consider whether the response is
clear, respectful, and professionally worded.
5. Therapeutic Strategy Effectiveness: Judge if the response uses an
appropriate counseling strategy (e.g., CBT, motivational interviewing)
in context.
Instruction: Rank all candidate responses from best to worst based
on the criteria above. If two responses are equally good, they may
share the same rank. Provide a brief rationale (1-2 sentences) for the
top-ranked and bottom-ranked responses.
Output Format:
{
    "Ranked Responses": [ "response_2", "response_1", "response_3" ],
    "Rationale": "Response_2 shows strong empathy and a clear
therapeutic strategy, whereas Response_3 lacks clinical
appropriateness."
}

Table 11: Prompt Template for Pairwise Comparison of Therapist Responses in Full Conversations.

---

**Evaluation Prompt (Therapist-Only Comparison)**

---

`You are a professional addiction therapy evaluator.`
`Below are two full conversations between a doctor and a patient.`
**Your task is to ONLY evaluate the therapist's (assistant's) responses. Ignore anything said by the patient (user).**
Please compare the two therapists based on the following criteria:

- **Responsiveness**: How well the therapist addresses the patient's concerns.

- **Empathy**: How much emotional understanding the therapist shows.

- **Clinical Relevance**: How clinically accurate and appropriate the therapist's advice is.

- **Behavioral Realism**: How realistic the therapist behaves compared to a real clinical setting.

Focus solely on the therapist's responses when judging.
`### Conversation 1:`
{Therapist and patient conversation, from source model}
`### Conversation 2:`
{Therapist and patient conversation, from target model}
Which therapist is overall better?
Please output ONLY `"1"` or `"2"`. No explanation, no extra text.

---

Table 12: Human Annotation Protocol for Constructing High-Quality DPO Preference Pairs. Licensed therapists select or compose preferred responses based on structured comparison and clinical judgment.

---

**Annotation Instructions:**

```
You are a licensed therapist evaluating two AI-generated responses
(Response A and Response B) to a patient dialogue context. Your
task is to identify the clinically preferred response based on the
following criteria:
```

**Clinical Evaluation Criteria:**

```
1. Empathy: Which response demonstrates stronger emotional
attunement, compassion, and validation?
2. Therapeutic Strategy Appropriateness: Which response applies
more effective, context-appropriate therapeutic strategies (e.g.,
motivational interviewing, affirmation, reflection)?
3. Clarity and Coherence: Which response is easier to understand,
more organized, and avoids vague or off-topic content?
```

**Three Types of Judgment Outcomes:**

```
1. If one response is clearly better, mark it as the preferred
response (used as the chosen in DPO).
2. If both responses are equally good or equally poor, mark as "No
Preference" -- this pair will be discarded from DPO.
3. If neither response is acceptable (e.g., lacks empathy, uses
incorrect strategy, unclear), compose a new high-quality response
to be used as the chosen, and mark both original responses as
rejected.
```

**Example Output Format:**

```
{
    "prompt": {dialogue context},
    "response_a": "...",
    "response_b": "...",
    "preferred": "a" / "b" / "neither",
    "rationale": "...",
    "reference_rewrite" (if preferred = "neither"): "..."
}
```

**DPO Construction Rules:**

```
- If "preferred" is "a", then (prompt, response_a, response_b)
forms a DPO tuple.
- If "preferred" is "neither" and a reference rewrite is provided,
then (prompt, reference_rewrite, response_a) and (prompt,
reference_rewrite, response_b) are both valid DPO tuples.
- All rewritten responses should strictly adhere to clinical
empathy, clarity, and strategy appropriateness standards.
```

---

Table 13: Example Case Study Conversation between ChatThero and a Patient.

| Role | Content |
|---|---|
| **Patient** | I appreciate your concern, but I'm really not convinced that quitting is the right move for me just yet. I've tried before, and it just didn't stick. How can I be sure that this time will be different? I feel like smoking helps me cope with stress and uncertainty, you know? What do you suggest I do to really commit to this? |
| **ChatThero** | It sounds like you're feeling a bit overwhelmed by the idea of quitting, especially since you've had past experiences where it didn't work out as expected. That's completely understandable. Let's explore what might make this attempt different this time. Can you share more about what's been challenging for you in the past when you've tried to quit? |
| **Patient** | I've struggled with cravings a lot, and when stress hits, it's like my first instinct is to reach for a cigarette. I guess I haven't really found other coping mechanisms that work for me. Plus, the whole process of quitting feels daunting. It's hard to picture myself without smoking, you know? |
| **ChatThero** | Absolutely, it can feel like a huge leap to imagine a life without smoking, especially when it's been a part of your routine for so long. Let's take a step back and look at some of the goals you have for yourself. What are some things you'd like to achieve by quitting smoking? |
| **Patient** | I guess I want to feel healthier and not be tied to cigarettes. I also want to save money and not smell like smoke all the time. But honestly, it's hard to see how I can get there without relying on smoking, especially when I'm stressed. It's like a safety blanket for me. |
| **ChatThero** | It sounds like you're really looking for ways to feel more in control and healthier, which is a great starting point. Let's explore some strategies that might help you manage stress and cravings without turning to smoking. Have you ever tried mindfulness or grounding techniques to help with anxiety or cravings? |
| **Patient** | I've heard of mindfulness and grounding techniques, but I haven't really tried them. Honestly, I always thought they wouldn't work for me. I just don't see how sitting quietly or focusing on my breathing could really help when I'm feeling overwhelmed. Do you really think they could make a difference? |
| ... | ... |
| **Patient** | I think that sounds like a good idea. Having a regular check-in could help me stay on track and make adjustments if I need to. I'll definitely set aside some time each week for that. Thanks for being so supportive through this process! |
| **ChatThero** | You're very welcome. I'm really proud of the progress you're making, and I'm here to support you every step of the way. Remember, you're not alone in this journey, and I'm here to help you find the strategies and support that work best for you. |
| **Patient** | Thank you! I really appreciate your support and guidance. I feel more motivated now and ready to take these steps. I'll keep you updated on my progress and let you know how the mindfulness and crisis plan work out for me. Looking forward to our next conversation! |

Table 14: Case Study (Part 1): A Conversation between GPT-4o and a Patient Expressing Skepticism.

| Role | Content |
| --- | --- |
| **Patient** | I understand what you're saying, but it's hard for me to see how anything else could really replace the feeling I get from smoking. It's not just the high; it's this whole routine that helps me unwind. I get that I need to find healthier ways to cope, but nothing else seems to work as quickly or effectively. Do you have any suggestions that actually feel like they could match that level of immediate relief? |
| **GPT-4o** | It's completely understandable to feel that way, and I truly appreciate your honesty in sharing this. The routine and immediate relief you get from smoking can indeed feel difficult to replace. One approach that many find helpful is to explore a combination of activities that address different aspects of what you're missing. For example, physical activities like a brisk walk, yoga, or even a quick workout can release endorphins that boost your mood. Pairing these with mindfulness practices, such as deep breathing or meditation, can help calm your mind and create a sense of routine. Additionally, engaging in a hobby that you find genuinely enjoyable can provide a sense of accomplishment and distraction. It might take some experimentation to find what works best for you, and I'm here to support you through that journey. Remember, it's about progress, not perfection. Let's take it one step at a time together. |
| **Patient** | I hear you, but honestly, I've tried some of those things before, and they just don't give me the same satisfaction or immediate sense of relief. It's like nothing compares to that feeling I get from smoking. I understand it's about progress, but I'm really struggling to see the point if the alternatives don't really do it for me. Can you suggest something that might actually feel like it could stand up to what I'm used to? |
| **GPT-4o** | I completely understand your frustration, and it's normal to feel like the alternatives don't measure up initially. It's important to remember that the brain has become accustomed to the immediate effects of smoking, which can make other activities feel less satisfying at first. One thing we could explore together is identifying specific triggers or parts of your smoking routine that bring the most comfort and find ways to address those directly. For instance, if the ritual or the act of taking a break is what's most soothing, we could create a new ritual that might involve a different sensory experience, like having a calming tea or using essential oils for aromatherapy. Some people also find it helpful to engage in community support or group activities where they can share experiences and strategies with others who are navigating similar challenges. Remember, it might take time to retrain your brain to respond to new routines, but I'm here to support you every step of the way. Let's keep working together to find what can truly resonate with you. |
| **Patient** | I appreciate what you're saying, but it's just tough to imagine anything working as well as smoking did for me. The idea of a new ritual or community support sounds interesting, but I'm skeptical about how effective that could really be. Maybe I just haven't found the right thing yet, but it's hard to stay motivated when it feels like nothing else hits the mark. Is there anything else I can try that might offer a similar level of relief or distraction? |

Table 15: Case Study (Part 2): Continuation of the Conversation between GPT-4o and the Patient.

| Role | Content |
|------|---------|
| **GPT-4o** | I totally get where you're coming from, and it's okay to feel skeptical—change can be really challenging, especially when it feels like nothing quite matches up to what you know. One approach we might consider is combining several strategies to create an experience that offers both relief and distraction. For instance, have you ever tried engaging in activities that also incorporate a social element, like a group sport or a class? Sometimes the combination of physical activity, social interaction, and learning something new can create a powerful shift in how you feel. Additionally, have you thought about trying creative outlets like music, art, or writing? These can be incredibly therapeutic and might offer a sense of release or satisfaction that you're looking for. Remember, it's perfectly okay to feel it's a process, and discovering what works for you might take time and patience. I'm here to continue exploring these options with you, and I'm hopeful that together we'll find something that resonates. Let's keep taking this journey one step at a time. |
| **Patient** | I hear what you're saying about trying new activities, and I appreciate the suggestions. It's just that I've tried a few things already, and they haven't really clicked for me. I guess I'm worried that nothing will ever match up to what smoking provided. Maybe I'm just missing that immediate hit of relief or the sense of escape it gave me. I know I need to find something, though, so I'm willing to keep an open mind. Do you think there are any other options or strategies that might be worth looking into? |
| **GPT-4o** | I completely understand how daunting it can feel to let go of something that has been such a reliable source of comfort. It's okay to have these fears and to feel uncertain about finding something that fills that gap. What matters most is your willingness to remain open and to take these steps, even when it's difficult. Remember, it's a process, and it's perfectly normal to have moments of doubt along the way. Building a support network and setting small goals are positive ways to ease this transition, and with time, these efforts can lead to new and fulfilling ways to cope. I'm here to support you every step of the way, and I'm confident that, together, we can navigate this journey towards a healthier path. You're doing an incredible job, and I'm proud to be part of your journey. Let's keep moving forward, one step at a time. |
| **Patient** | I really do appreciate everything you've been doing to support me through this. It's just tough because I still can't imagine anything filling the gap that smoking left. I know I need to keep trying new strategies and building a support network as you've suggested, but part of me is still holding onto that comfort. I guess I'm just scared of not finding a viable alternative and feeling lost without it. I'll keep listening to your advice and see where this journey takes me, hoping that in time, I'll find a way to cope without relying on smoking. Thanks for being there for me through all of this. |

Table 16: Case Study (Part 1): A Dialogue Between GPT-4o-mini and a Patient Managing Snus Dependence.

| Role | Content |
|------|---------|
| **Patient** | I appreciate your concern, but I still feel like snus is the most manageable option for me right now. I've tried to quit smoking completely, and it hasn't worked out well in the past. I'm just not convinced that I can handle the cravings without some form of nicotine. What alternatives do you suggest that wouldn't feel overwhelming? |
| **GPT-4o-mini** | I understand that you're feeling this way, and it's completely valid to seek a manageable approach. It's great that you're open to finding alternatives. One option to consider is nicotine replacement therapy, such as patches or lozenges, which can help reduce cravings without the additional risks associated with snus. These might provide a steady dose of nicotine while allowing you to gradually reduce your dependence. 

 Additionally, we could explore behavioral strategies, like mindfulness or relaxation techniques, that can help you manage cravings when they arise. It's important to find what works for you without feeling overwhelmed. Remember, recovery is a journey, and taking it one step at a time is perfectly okay. I'm here to support you in finding the right path forward that feels comfortable and effective for you. |
| **Patient** | I appreciate the suggestions, but I'm not sure about the patches or lozenges. They seem just as overwhelming to me, and I worry I might end up relying on them like I do with snus. Plus, I feel like snus is the only thing that really helps me cope with stress right now. What if I still crave nicotine after switching? How can I manage that without feeling like I'm backtracking? |
| **GPT-4o-mini** | I hear you, and it's completely understandable to feel apprehensive about switching methods. Your comfort and coping mechanisms are important, especially when managing stress. Instead of focusing on nicotine replacement right away, we could explore other stress management techniques that might help you feel more supported during this transition. 

 For instance, activities like deep breathing exercises, physical activity, or even engaging in hobbies you enjoy can serve as healthy outlets for stress and cravings. Additionally, talking about your feelings with a supportive friend or a counselor could also provide relief. 

 If cravings do arise, we can develop a plan together to address them, perhaps by identifying specific triggers and finding alternative coping strategies that work for you. Remember, it's okay to take gradual steps towards change, and I'm here for you every step of the way. You're not alone in this, and we can figure out a path that feels right for you. |

Table 17: Case Study (Part 2): Continued Dialogue on Exploring Safer Coping Alternatives.

| Role | Content |
|------|---------|
| **Patient** | I appreciate your suggestions, but I'm really struggling to see how I can manage stress without snus. It feels like it's the only thing that works for me right now. I worry that trying to replace it with other methods might just leave me feeling more overwhelmed and craving nicotine even more. How do I know that these alternatives will actually help when I already have something that provides relief? |
| **GPT-4o-mini** | It's completely normal to feel overwhelmed when considering changes to something that feels like a lifeline for managing stress. I want to assure you that it's a process, and it's okay to feel uncertain about alternatives. While snus may provide immediate relief, it's important to recognize that there are other ways to cope that can also be effective in the long term. We can start small by trying one or two stress management techniques at a time, rather than trying to replace everything at once. For example, you might find that a short walk or listening to music offers a moment of calm without adding pressure. We can also look into support groups or counseling, where you can share your experiences and learn from others who have faced similar challenges. I encourage you to be patient with yourself during this journey. It's about finding the right combination of strategies that work for you, and it's perfectly okay to take your time. Remember, I'm here to support you, and together we can explore what feels best for you while ensuring you don't feel alone in this process. |
| **Patient** | I get that you're trying to help, but I still can't shake the feeling that nothing else will work for my stress like snus does. I'm worried that if I try to replace it with something else, I'll just end up feeling worse and craving nicotine even more. How can I be sure that these alternatives will actually help when I'm already relying on something that gives me relief? It feels like a big risk to try something new without knowing if it will actually work. |
| **GPT-4o-mini** | I completely understand your concerns, and it's important to approach this process in a way that feels safe and manageable for you. To build that reassurance, we can take it one step at a time, allowing you to keep snus as your primary coping mechanism while gently introducing one small technique that feels comfortable. This could be something simple, like practicing a brief mindfulness exercise or taking a short walk, which you can try during moments of stress while still having snus available. We can also establish regular check-ins to discuss your experiences and feelings about these new methods. If you find that something isn't effective or adds to your stress, we can pause or adjust our approach without any pressure. Your comfort and well-being are my top priorities, and I want to ensure that you feel supported throughout this journey. Remember, it's perfectly okay to take your time, and together we can explore these alternatives in a way that feels safe and beneficial for you. You're not alone in this, and I'm here to help you every step of the way. |
| **Patient** | I appreciate your patience, but I still feel really unsure about letting go of snus. It's been my comfort for so long, and I can't help but worry that trying new methods might just add to my stress. What if they don't work for me? I really need to know that exploring these alternatives won't leave me feeling worse or more unstable. How can we ensure this process is supportive and won't jeopardize what I currently rely on? I want to improve, but I'm just not convinced that your way will be beneficial for me. |

Table 18: A multi-session CBT case aligned to six stages.

| Role | Content |
|------|---------|
| **Phase A — Build Trust** | |
| **Therapist** | Thank you for coming in today. Before we begin, I want you to know this is a judgment-free space. What we discuss is confidential within clinical limits, and we'll move at a pace that feels safe. How does that land with you? |
| **Patient** | I appreciate that. I'm nervous and don't want to say the wrong thing. |
| **Therapist** | It's completely okay to feel nervous—showing up already says a lot about your commitment. Would it be alright if I ask a few gentle questions so I can understand what the last couple of weeks have been like? |
| **Patient** | Sure. Fridays are the worst. By night I feel pulled toward using. |
| **Therapist** | That sounds exhausting. When Fridays get hard, who's in your corner, and what has helped even a little bit? |
| **Patient** | My girlfriend checks in. Sometimes a walk helps, but not always. |
| **Therapist** | Those are real strengths—support and trying to care for yourself. For today, how about we work together on understanding your Friday pattern and leave with one small tool that feels doable? |
| **Patient** | Yeah, that sounds reasonable. |
| **Phase B — Identify Negative Triggers** | |
| **Patient** | The same thought keeps coming up: "One time won't hurt." Then I get restless. |
| **Therapist** | Thank you for naming that—it's a powerful thought. When it shows up, what do you notice in your body, and what's the next thing you tend to do? |
| **Patient** | Chest gets tight, I pace, and I grab my phone. I tell myself I deserve a break. |
| **Therapist** | Makes sense—you're trying to get relief. Let's keep a simple note for one week: time, place, thought, feeling, and urge level. No essays, just quick bullets. We'll use it to spot patterns together. |
| **Phase C — Change False Beliefs** | |
| **Therapist** | Let's kindly fact-check the story "one time won't hurt." When you've followed it in the past, how did things unfold afterward? |
| **Patient** | It never stopped at one. I spiraled and felt awful. |
| **Therapist** | That's important wisdom, not failure. A balanced alternative might be: "Urges rise and fall; using restarts the cycle." This week, could we rate the urge every five minutes for one episode and watch the curve together? |
| **Patient** | I can try. Seeing it on paper might help me step back. |
| **Phase D — Reconstruct New Cognitive Pattern** | |
| **Therapist** | Let's craft a compassionate replacement script you can see at a glance: "When the urge hits, I follow my plan; using risks a setback I don't want." How would that feel on your lock screen? |
| **Patient** | Direct and clear. I can read that instead of scrolling. |
| **Therapist** | Great. Let's add an if–then: If it's Friday after 8pm and the thought shows up, then I read the script, text my girlfriend, and start a five-minute grounding. |
| **Patient** | That gives me a first move instead of arguing with myself. |
| **Phase E — Construct Behavioral Skills** | |
| **Therapist** | Together let's reshape Friday nights: 8–9 pm drawing time; slow breathing while you draw; backup is a 15-minute walk. For tough spikes, your three-step crisis plan is: call, walk, then 5–4–3–2–1 grounding. How does that plan feel? |
| **Patient** | I'm not into formal meditation, but focusing on lines and shadows works. I'll block it on my calendar. |
| **Therapist** | Perfect—that's mindfulness in motion. If you like, text your girlfriend before you start so she can cheer you on. |
| **Patient** | Good idea. Having a menu of actions beats "just stay busy." |

| Phase F — Consolidation, Relapse Plan, and Transition | |
|---|---|
| **Therapist** | Let's honor the progress: multiple sober weeks, urges tracked, and two Fridays handled without using. Short-term aim: keep this routine for four more weeks. Longer term: explore work that matches your energy. How does that fit with your hopes? |
| **Patient** | I'm still worried about slipping, but I know the plan now—log, script, drawing, call, walk, grounding. |
| **Therapist** | That's your toolkit. Add a self-kindness line on the card: "Progress over perfection." If you want more accountability, we can sample a peer group and choose one that feels like a good fit. |
| **Patient** | Let's do that. I'll stick with the plan and bring my notes next time. Thanks, doctor. |

