# OpenReview forum: "ChatThero: A Language Agent for Recovery Support"
_ICLR.cc/2026/Conference — ICLR 2026 Conference Withdrawn Submission_

### Official Review · Reviewer_kBgi · 2025-10-30

**Soundness:** 3
**Presentation:** 2
**Contribution:** 2
**Rating:** 4
**Confidence:** 3

**Summary:**

This paper introduces ChatThero, a multi-session language model designed to support addiction recovery (Substance Use Disorder, SUD). It integrates principles from Motivational Interviewing (MI) and Cognitive Behavioral Therapy (CBT) within a simulated multi-agent environment that mimics real-world stressors, relapses, and patient resistance. By training through repeated therapy-like interactions, ChatThero learns to adapt its strategies across sessions. Experimental results show that it outperforms general-purpose large language models (such as GPT-4o) in motivating patients, expressing empathy, and maintaining therapeutic consistency across sessions.

**Strengths:**

The paper introduces a novel multi-agent, multi-session training framework that captures long-term dialogue continuity and adaptive strategy use.

The model achieves higher empathy scores, more realistic behavioral patterns, and faster motivational progress compared to baseline models, demonstrating its specialized capability in recovery-related dialogue.

ChatThero explicitly encodes principles from MI and CBT, giving its responses stronger psychological grounding and interpretability from a therapeutic perspective.

**Weaknesses:**

Although ChatThero performs well in experiments with “virtual patients,” those patients were themselves generated by language models, and the evaluators were often GPT-4o or human raters. This “model evaluating a model” design creates a circular validation problem and lacks truly independent, objective assessment

The simulated environment is tightly controlled, but there is no data from real recovery participants or longitudinal follow-up. Without observing actual user responses, adherence, or emotional fluctuations, the generalizability of the results to real clinical settings remains limited.

Real addiction recovery involves rich multimodal cues such as tone, voice, and body language. ChatThero operates purely on text, which limits its ability to perceive affective nuances and adjust its interventions dynamically.

Although the authors claim to integrate Motivational Interviewing (MI) and Cognitive Behavioral Therapy (CBT), they do not clearly show how the model’s internal strategy mapping corresponds to concrete therapeutic actions (e.g., reflective listening, cognitive reframing). It feels more like semantic imitation than genuine internalization of psychological mechanisms.

**Questions:**

N/A

---

### Official Review · Reviewer_12c5 · 2025-10-31

**Soundness:** 3
**Presentation:** 3
**Contribution:** 3
**Rating:** 4
**Confidence:** 4

**Summary:**

This work introduces ChatThero, a low-cost, memory-persistent, and stressor-aware autonomous language agent designed to support long-term addiction recovery. Unlike prior approaches that fine-tune LLMs on therapist–patient dialogues, ChatThero is trained in a multiagent simulated therapy environment with dynamic stressors and clinically grounded strategies such as motivational interviewing and cognitive behavioral therapy. Evaluations by clinicians and LLM judges show that ChatThero outperforms GPT-5 in empathy, clinical relevance, and behavioral change, significantly boosting users’ motivation and confidence while requiring fewer dialogue turns with difficult patients.

**Strengths:**

1. I like the dynamic changing mental state setting.
2. I like the multi-session setting.

**Weaknesses:**

1. The paper generates lots of patient profiles via prompting. However, there is no human evaluation or other kinds of comparison with real patient cases. I'm not sure if those synthetic profiles are reasonable.
2. In sec 4.2, the decoding temperature is 0.7. The model performance can be influenced by temperature when the data samples are not many.
3. For human evaluation, can you provide more information on human annotator? how many data samples were evaluated?

**Questions:**

See weaknesses.

---

### Official Review · Reviewer_QF7q · 2025-11-01

**Soundness:** 3
**Presentation:** 2
**Contribution:** 2
**Rating:** 4
**Confidence:** 3

**Summary:**

This paper introduces ChatThero, a multi-agent large language model framework designed to generate responses for long-term behavior change and therapeutic support in addiction recovery. The system comprises several key components: a patient agent with a structured profile including personality traits, an environment agent that simulates stressors between sessions, and a therapy agent equipped with multiple toolkits and strategies grounded in motivational interviewing and cognitive behavioral therapy. The framework also incorporates a multi-session episode structure that adapts therapeutic strategies over time. For experimental validation, the authors construct a synthetic dataset initialized from Reddit posts and generated through the multi-agent system. Evaluation employs two approaches: LLM-as-a-judge to assess response motivation and confidence, and human evaluation across five dimensions, including responsiveness and empathy. Additionally, the authors measure time-to-success across multi-session episodes. Experiment results show that ChatThero outperforms single LLMs such as GPT-4 in both automatic and human evaluations. Interestingly, the authors also find that employing more therapeutic strategies does not necessarily improve response quality.

**Strengths:**

The primary strength of this work lies in the demonstrated effectiveness of the multi-agent architecture for generating contextually appropriate responses that support long-term behavior change. By grounding the system design in established psychological and therapeutic theories, specifically motivational interviewing and cognitive behavioral therapy, the authors successfully demonstrate how these evidence-based approaches can be operationalized within a multi-agent framework.

**Weaknesses:**

Several significant concerns merit attention. First, the claimed efficiency advantage requires empirical substantiation. While the authors claim that ChatThero is an “innovative low-cost” solution, the multi-agent architecture is considerably more complex than a single LLM baseline. To support the efficiency claims, the authors should provide concrete metrics such as computational cost, inference time, or API call counts comparing ChatThero against single-LLM alternatives.

Second, the experimental design raises concerns about evaluation validity. The synthetic dataset, though initialized from Reddit, is constructed following the multi-agent simulation approach described in lines 296-297. This creates a potential circularity. ChatThero is essentially evaluated on data generated using principles embedded in its own architecture, which may inflate performance metrics. While the authors acknowledge the lack of suitable existing datasets and the necessity of synthetic data, this limitation necessitates more rigorous internal validation. Comprehensive ablation studies would strengthen the evaluation, for instance, by systematically removing components from each agent type or by evaluating on datasets constructed with alternative assumptions about patient profiles or therapeutic approaches.

**Questions:**

- Which specific therapeutic strategies prove most beneficial, and which are less effective? This analysis would provide actionable insights for both the NLP community and practitioners.
- "Real-world" model in Table 3. Is it a model or a dataset? If it refers to a dataset with human-annotated therapeutic interactions, why was it not used more extensively for evaluation? If human-validated data exists, direct comparison would significantly strengthen the empirical claims.
- The text and items in the figures are difficult to read. Improving font sizes and visual clarity would enhance accessibility.

---

### Official Review · Reviewer_7Hds · 2025-11-07

**Soundness:** 3
**Presentation:** 2
**Contribution:** 3
**Rating:** 2
**Confidence:** 4

**Summary:**

The paper introduces ChatThero, a domain-aligned multi-session, stressor-aware, memory-persistent language agent for addiction recovery support. The system uses a three-agent simulation (Patient, Environment, Therapy) to model realistic treatment dynamics, with patient profiles synthesized from anonymized recovery-forum narratives, explicit between-session stressors, and a therapy policy that sequences MI/CBT/harm-reduction tactics across a six-stage plan (S1–S6). Training is two-stage: SFT on clinician-guided synthetic dialogues followed by DPO that prefers better tactic timing and carryover. Evaluation mixes an LLM-as-judge with blinded clinician ratings and multi-session outcome metrics (motivation, confidence, time-to-success). Reported gains include higher empathy/clinical relevance and faster progress on medium/hard cases versus strong baselines.

**Strengths:**

•	Clear, therapy-process framing. The MI/CBT layering with explicit OARS tools and six CBT stages is well-operationalized; Figure 1 (p. 2) makes the control flow and cross-session memory/stressor interactions concrete.

•	Training recipe matches the task. SFT establishes safe MI/CBT scaffolding; DPO targets strategy selection and timing (e.g., probe vs plan, switch under stressors).

•	Multi-session evaluation and metrics. The work looks beyond single-turn quality to trajectories of motivation/confidence and time-to-success, with ablations and clinician/LLM agreement.

**Weaknesses:**

•	Heavy reliance on synthetic data and LLM evaluators. Although the paper includes blinded clinicians, much of the training data and judging pipeline are LLM-generated/LLM-scored; the human study involves two licensed physicians, which may limit robustness and generalizability of ratings.

•	External validity and cultural generalization. Authors note English-only, Western-context scenarios and absence of longitudinal real-world outcomes; deployment readiness remains unclear.


•	Baseline fairness and ceilings. Some result narratives describe “ceiling effects” for easy cases and modest margins elsewhere; clearer, fully standardized prompts/decoding and ablation vs. therapy-tooling alone would help isolate where gains truly come from.

•	Outcome definitions and success criteria. Motivation/confidence are clinically meaningful, but their operationalization by an LLM judge may introduce bias; more extensive human calibration and inter-rater stats in-paper would strengthen claims.

**Reproducibility:** The Reproducibility Statement references a code URL that is currently blank/inaccessible. As a result, the experiments are not reproducible at review time. Please provide an anonymous, working code package (zip or repo) and list the exact environment (framework/versions), data processing steps and splits, training/eval commands (with seeds), and hyperparameters. In the absence of these materials, I must assign a low reproducibility score, which also reduces my overall confidence in the empirical claims. (ICLR encourages providing such materials for reproducibility.)

**Questions:**

1.	Human evaluation scale & agreement. Beyond the appendix, can you report inter-rater reliability (e.g., Cohen’s κ/ICC) for the two clinicians and alignment vs. the LLM judge on the main test set (not just a sample)?

2.	Stressors & ledger validity. How were stressor types/severities validated clinically? Any user studies showing that ledger updates produce realistic relapse dynamics beyond simulated agents?

3.	Ablations on therapy primitives. What happens if you remove specific MI/CBT tools (e.g., decisional balance, refusal-skills rehearsal)? Which primitives contribute most to gains in medium/hard settings?

4.	Generalization & safety. Any evidence the policy transfers to non-English or to comorbid presentations (e.g., anxiety, trauma) without retuning? What additional guardrails are in place for crisis scenarios beyond rule-based filters?

---

### Note · Authors · 2026-01-06

I have read and agree with the venue's withdrawal policy on behalf of myself and my co-authors.